# QUERY-CENTRIC GRAPH RETRIEVAL AUGMENTED GENERATION

## ABSTRACT

Graph-based retrieval-augmented generation (RAG) enriches large language models (LLMs) with external knowledge for long-context understanding and multi-hop reasoning, but existing methods face a granularity dilemma: fine-grained entity-level graphs incur high token costs and lose context, while coarse document-level graphs fail to capture nuanced relations. We introduce QCG-RAG, a query-centric graph RAG framework that enables query-granular indexing and multi-hop chunk retrieval. Our query-centric approach leverages Doc2Query and Doc2Query-- to construct query-centric graphs with controllable granularity, improving graph quality and interpretability. A tailored multi-hop retrieval mechanism then selects relevant chunks via the generated queries. Experiments on LiHuaWorld and MultiHop-RAG show that QCG-RAG consistently outperforms prior chunk-based and graph-based RAG methods in question answering accuracy, establishing a new paradigm for multi-hop reasoning.

## 1 INTRODUCTION

Retrieval-augmented generation (RAG) has become a standard approach for improving the factuality of large language models (LLMs) by grounding them in external knowledge (Gao et al., 2023). By retrieving supporting chunk-based evidence, RAG not only enhances accuracy but also mitigates hallucinations common in LLM generation (Yu et al., 2024). However, existing chunk-based RAG pipelines often fail when queries are underspecified or require multi-hop reasoning, due to a mismatch between query intent and the fragmented distribution of supporting chunk-level evidence in retrieval (Tang & Yang, 2024).

Graph-based RAG provides a more effective strategy for capturing relational knowledge and associating dispersed evidence across documents (Peng et al., 2024; Han et al., 2024; Yu et al., 2025) compared to chunk-based RAG. In particular, GraphRAG (Edge et al., 2024a) constructs a fine-grained entity-centric knowledge graph (KG) with community summaries, allowing retrieval to exploit both text similarity and entity-level connections. This fine-grained KG strengthens semantic coherence across dispersed entity-level evidence, thereby enabling deeper contextual understanding. GraphRAG is especially effective for multi-hop reasoning and long context understanding, enabling applications in question answering over massive documents (Ghassel et al., 2025; Chen et al., 2024) and long dialogues (Fan et al., 2025; Zhang et al., 2025b).

However, existing graph-based RAG methods face a fundamental trade-off: fine-grained entity-level graphs incur prohibitive token costs and often lose semantic coherence, while coarse-grained document-level graphs sacrifice nuanced relations. For instance, GraphRAG imposes considerable computational overhead: constructing fine-grained entity-level graphs and producing community summaries inflate token budgets and inference cost, and can erode contextual coherence (Min et al., 2025; Zhang et al., 2025a). Recent work simplifies graph construction by skipping community detection and summarization, and instead using lightweight information extractors (e.g., SpaCy (Honnibal et al., 2020)) or fine-tuned small LLMs (e.g., Phi-3-3.8B (Abdin et al., 2024), but at the cost of lower graph quality. Other efforts adopt hierarchical index graphs with document-level links (Chen et al., 2024), which are efficient but too coarse to support nuanced reasoning. These limitations highlight the need for strategies that balance granularity and effectiveness in graph-based RAG.

Document expansion techniques, such as Doc2Query (Nogueira et al., 2019) for generating queries from documents and Doc2Query-- (Gospodinov et al., 2023) for filtering irrelevant queries, pro-

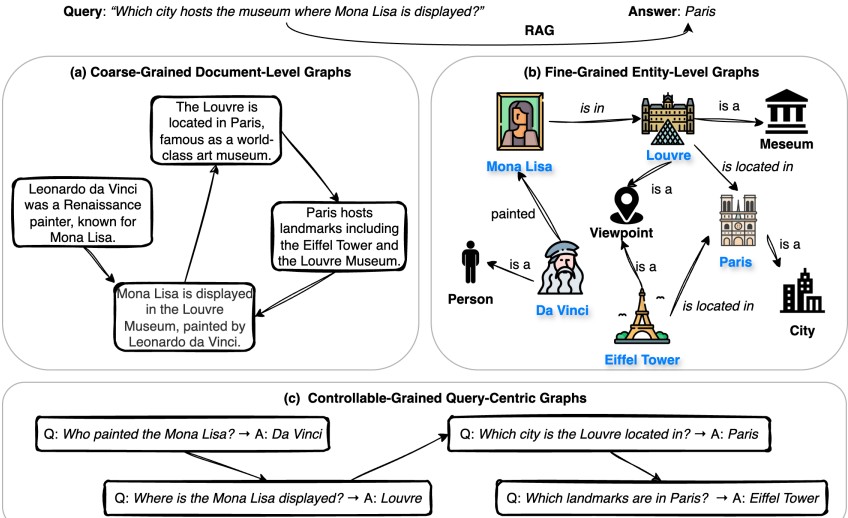

Figure 1: Illustration of (a) coarse-grained document-level graphs, (b) fine-grained entity-level graphs, and (c) controllable-grained query-centric graphs for RAG.

vide a query-centric expansion paradigm. We argue that integrating Doc2Query-generated queries with graphs enables the construction of a controllable query-centric graph that lies between fine-grained entity-level and coarse-grained document-level representations. To overcome the above mentioned granularity dilemma, we propose Query-Centric Graph Retrieval-Augmented Generation (QCG-RAG), a framework that tailors graph construction and retrieval to the granularity of queries. QCG-RAG introduces a query-centric graph (QCG), constructed with Doc2Query and Doc2Query-- techniques to generate query-guided nodes and edges, yielding indexing graphs with controllable granularity. On top of this graph, we design a multi-hop retrieval & generation mechanism that efficiently selects relevant chunks via generated queries, enabling precise reasoning across dispersed evidence. Figure 1 illustrates (a) coarse-grained document-level graphs, (b) fine-grained entity-level graphs, and (c) controllable-grained query-centric graphs for RAG. The main contributions of this paper are threefold:

• **Query-Centric Graph Construction.** We present the first query-driven, granularity-controllable graph construction framework for RAG, which improves graph quality and preserves interpretability via generated queries.

• **Query-Centric Retrieval & Generation Mechanism.** We develop an effective multi-hop retrieval process on QCG, which enables accurate retrieval of relevant text chunks for complex queries.

• **Comprehensive Evaluation.** We conduct experiments demonstrating that QCG-RAG consistently outperforms prior chunk-based and graph-based RAG methods in question answering accuracy.

## 2 RELATED WORK

**Chunk-based RAG.** Conventional chunk-based RAG frameworks, such as Naive RAG (Gao et al., 2023), typically follow a four-step pipeline: (1) Chunking: documents are segmented into fixed-length units using sliding windows or semantic boundary detection to balance granularity and context; (2) Embedding: chunks are encoded into dense vectors with pretrained encoders (e.g., BGE-M3 (Multi-Granularity, 2024) and Sentence-BERT (Reimers & Gurevych, 2019)) and indexed in vector databases (e.g., FAISS (Douze et al., 2024), Milvus (Wang et al., 2021), and ElasticSearch[1]); (3) Retrieval: user queries are embedded and compared against chunk vectors via similarity metrics, optionally reranked with cross-encoders; and (4) Generation: top-k retrieved chunks are concatenated with the query and fed into LLMs (e.g., GPT-4 (Achiam et al., 2023), Qwen2.5 (Qwen et al., 2025), DeepSeek-R1 (Guo et al., 2025)) to produce responses. While effective for factual grounding, this pipeline often suffers from semantic misalignment between retrieval and generation, as well as insufficient coverage for multi-hop queries. These limitations motivate graph-based RAG approaches, which introduce structured semantic representations to better bridge the gap between queries and dispersed evidence.

---

[1] https://github.com/elastic/elasticsearch

**Graph-based RAG.** Graph-based RAG extends conventional chunk-based RAG by introducing structured knowledge graphs that capture entity-level relations, enabling retrieval not only by text similarity but also through graph-based reasoning (Peng et al., 2024). Typical implementations, such as GraphRAG (Edge et al., 2024a), construct entity-centric graphs using LLM-based extraction, cluster entities into communities, and generate community summaries for retrieval alongside original documents, thereby supporting cross-chunk association and multi-hop reasoning. This line of work has motivated both (1) *simplified variants* (e.g., Fast GraphRAG[2], LightRAG (Guo et al., 2024), LazyGraphRAG (Edge et al., 2024b), Triplex[3], and E²GraphRAG (Zhao et al., 2025)) that reduce construction costs through model and pipeline optimization, and (2) *structural extensions* (e.g., KG-Retriever (Chen et al., 2024), Mixture-of-PageRanks (Alonso & Millidge, 2024)) that organize knowledge into hierarchical indexes to improve retrieval coverage, highlighting the persistent granularity trade-off in graph-based RAG.

In particular, Fast GraphRAG simplifies this pipeline by eliminating community detection and summary generation to reduce LLM usage. LightRAG further streamlines the process by removing the community component entirely, making the system more lightweight. LazyGraphRAG replaces LLM-based extraction with small local models that capture noun co-occurrences, while generating community summaries dynamically at query time. Triplex leverages a fine-tuned lightweight LLM (Phi3-3.8B) and E²GraphRAG employs traditional NLP tools such as SpaCy (Honnibal et al., 2020) for graph extraction, significantly lowering construction costs. Beyond simplification, KG-Retriever constructs a hierarchical index that integrates an entity-level knowledge graph with a document-level layer, enhancing intra- and inter-document connectivity to support efficient cross-document retrieval and multi-granularity access. To address the granularity dilemma of graph-based RAG, we propose Query-Centric Graph RAG (QCG-RAG), which enables controllable query-centric graph construction to balance granularity and align retrieved evidence with user intent.

**Document Expansion with Doc2Query & Doc2Query--.** Doc2Query (Nogueira et al., 2019) is a document expansion technique that trains sequence-to-sequence models (e.g., T5 (Raffel et al., 2020)) to generate queries likely associated with a given document, thereby improving retrieval by appending the generated queries to the document chunks. Doc2Query-- (Gospodinov et al., 2023) refines this approach by filtering out irrelevant or hallucinated queries based on their similarity to the original chunks, which improves retrieval quality while reducing index overhead. The queries produced by these methods naturally operate at an intermediate granularity: richer and more interpretable than fine-grained entity triples, yet more precise than coarse-grained document chunks. This property makes them well suited for integration with graph-based RAG, where query nodes enable controllable, query-centric graph (QCG) construction that balances granularity and enhances both the accuracy and interpretability of multi-hop retrieval.

## 3 METHODOLOGY

### 3.1 PRELIMINARIES

We consider the task of open-domain question answering (QA) under the retrieval-augmented generation (RAG) paradigm. Formally, let $\mathcal{C} = \{c_1, c_2, \ldots, c_N\}$ denote a collection of text chunks derived from a document corpus $\mathcal{D}$, where each chunk $c_i(i \in [1, N])$ is a contiguous segment of text (e.g., a dialogue or passage). Given a user query $q_u \in \mathcal{Q}$, the goal is to generate an answer $a$ by retrieving relevant chunks from $\mathcal{C}$ and conditioning a large language model (LLM) on both $q_u$ and the retrieved chunk evidence.

**Doc2Query.** Doc2Query (Nogueira et al., 2019) is a document expansion technique that enrich the retrieval space, where each chunk $c_i$ is used as a prompt to an LLM to generate multiple synthetic query:

$$\mathcal{Q}_{g,i} = \{q_{g,i}^1, q_{g,i}^2, \ldots, q_{g,i}^M\},$$

where $q_{g,i}^j(j \in [1, M])$ denotes a generated query that is grounded in the chunk content of $c_i$. $M$ is the number of generated queries per chunk $c_i$. This document expansion allows the retrieval module to associate user queries not only with explicit surface forms of chunks, but also with semantically aligned synthetic queries.

---

[2] https://github.com/circlemind-ai/fast-graphrag
[3] https://huggingface.co/SciPhi/Triplex

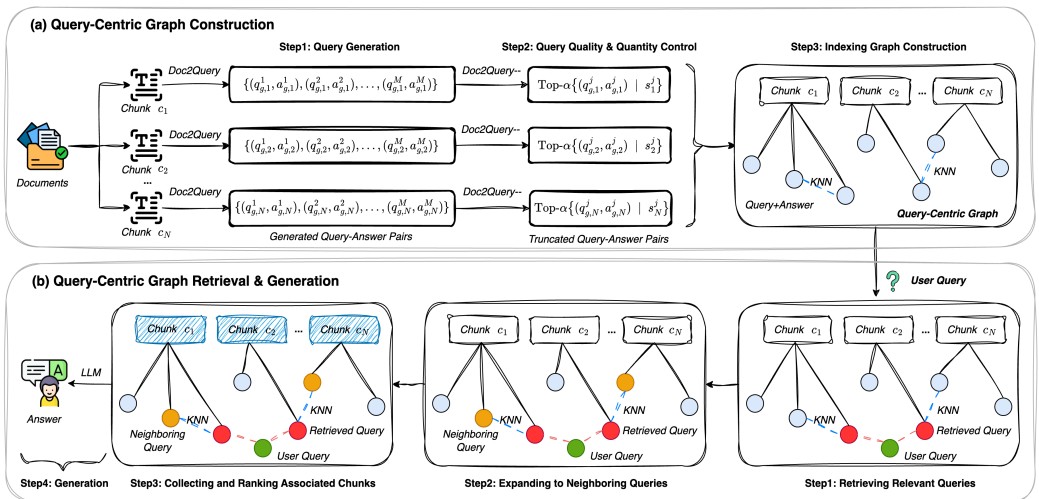

Figure 2: Overview of our proposed QCG-RAG framework, which consists of (a) Query-Centric Graph Construction and (b) Query-Centric Graph Retrieval & Generation.

**Doc2Query--.** Doc2Query-- (Gospodinov et al., 2023) is a further filtering step to mitigate noise introduced by synthetic queries. Specifically, the semantic similarity is computed between each generated query $q_{g,i}^j$ and its originating chunk $c_i$, denoted $\text{sim}(q_{g,i}^j, c_i)$. The generated queries are ranked by similarity and only the top $\alpha$-fraction (with $\alpha \in (0, 1]$) are retained:

$$\mathcal{Q}_{g,i}^\alpha = \text{Top-}\alpha\big\{q_{g,i}^j \mid \text{sim}(q_{g,i}^j, c_i)\big\}.$$

This maintains synthetic queries faithfulness to $c_i$ while improving retrieval precision.

**Standard RAG Pipeline.** In the RAG framework, the retrieval component returns a ranked list of top-$K$ chunks $\mathcal{C}_{\text{top-}K} = \{c_1, \ldots, c_K\}$ based on a similarity function $\text{sim}(q, c)$, often instantiated by dense embedding retrievers. The generation component then produces the final response:

$$a = \text{LLM}\big(q_u \mid \mathcal{C}_{\text{top-}K}\big).$$

We use this setting, augmented with extended Doc2Query and Doc2Query--, as the basis for introducing our proposed Query-Centric Graph Retrieval Augmented Generation (QCG-RAG) framework.

### 3.2 FRAMEWORK OVERVIEW

Prior graph-based RAG approaches exhibit inherent limitations in granularity: either too fine-grained with entity-level graphs or too coarse-grained with document-level graphs. To address this granularity dilemma, we propose **Query-Centric Graph Retrieval-Augmented Generation (QCG-RAG)**, a novel framework that integrates Query-Centric Graph (QCG) indexing and retrieval mechanisms to enhance response accuracy and interpretability. As illustrated in Figure 2, our framework consists of two major steps: Query-Centric Graph Construction and Query-Centric Graph Retrieval & Generation.

**Query-Centric Graph Construction.** We first enrich the retrieval space by generating synthetic query–answer pairs from each text chunk using extended Doc2Query. Given a chunk $c_i$, an LLM generates a set of queries $\{q_{g,i}^j\}$ with corresponding answers $\{a_{g,i}^j\}$ that faithfully represent the content of $c_i$. To reduce noise, we apply extended Doc2Query--, which ranks the generated query-answer pairs by their semantic similarity to the source chunk and retains only the top $\alpha$ fraction. Note that, unlike the original Doc2Query and Doc2Query--, which leverage queries only, the extended methods incorporate query–answer pairs. The resulting high-quality queries serve as nodes in the Query-Centric Graph (QCG), where edges represent *chunk–query* membership and *query–query* similarity relations. This process enables graph construction with controllable granularity while strengthening chunk–query associations and capturing richer semantic relations among queries.

**Query-Centric Graph Retrieval & Generation.** Given a user query $q_u$, retrieval proceeds in four steps: (1) retrieving semantically related queries from the graph (*Query → Retrieved Queries*); (2) expanding to neighboring queries to capture multi-hop relations (*Retrieved Queries → Neighboring*

*Queries*); (3) aggregating and ranking associated chunks linked to the retrieved query set (*Query Set → Chunk Set*); and (4) generating the final response by conditioning the LLM on the user query and the top-$K$ retrieved chunks (*Top-$K$ Chunk Set → Generation*). This query-centric retrieval design enables flexible granularity control, better coverage of relevant document segments, and interpretable reasoning paths.

### 3.3 QUERY-CENTRIC GRAPH CONSTRUCTION

This component builds a controllable-granularity *Query-Centric Graph* by leveraging and extending Doc2Query and Doc2Query--. It consists of three sequential steps: (1) query generation, (2) query quality and quantity control, and (3) indexing graph construction.

**Step 1: Query Generation.** Given a corpus of chunks $\mathcal{C} = \{c_1, c_2, \ldots, c_N\}$, each chunk $c_i$ is expanded into multiple synthetic query–answer pairs via extended Doc2Query:

$$\mathcal{Q}_{g,i} = \{(q_{g,i}^1, a_{g,i}^1), (q_{g,i}^2, a_{g,i}^2), \ldots, (q_{g,i}^M, a_{g,i}^M)\},$$

where $q_{g,i}^j (j \in [1, M])$ is a generated query grounded in $c_i$ and $a_{g,i}^j$ is its corresponding answer. Query–answer pairs enrich semantic signals by anchoring queries with contextualized answers, reducing ambiguity, strengthening chunk alignment, and expanding retrieval through multiple grounded entry points for each chunk.

**Step 2: Query Quality and Quantity Control.** For each generated query-answer pair $(q_{g,i}^j, a_{g,i}^j)$, we compute its semantic similarity with the source chunk $c_i$ according to extended Doc2Query--:

$$s_i^j = \text{sim}(q_{g,i}^j \oplus a_{g,i}^j, c_i),$$

where $\text{sim}(\cdot, \cdot)$ denotes the semantic similarity function (e.g., cosine similarity). $\oplus$ denotes text-level concatenation. The queries are ranked by $s_i^j$, and only the top $\alpha$ percentile are retained:

$$\mathcal{Q}_{g,i}^\alpha = \text{Top-}\alpha\{(q_{g,i}^j, a_{g,i}^j) \mid s_i^j\}, \quad \alpha \in (0, 1].$$

This mechanism maintains the fidelity of synthetic query-answer pairs to the source content $c_i$, which in turn contributes to more precise and reliable retrieval.

**Step 3: Indexing Graph Construction.** We construct a two-layer indexing graph $\mathcal{G} = (V, E)$ consisting of a query-level layer and a chunk-level layer:

$$V = \mathcal{C} \cup \mathcal{Q}_g, where \ \mathcal{Q}_g = \bigcup_{i=1}^{N} \mathcal{Q}_{g,i}^\alpha.$$

Edges $E$ capture two types of relations:

$$E_{\text{intra}} = \{(q, q') \mid q' \in \text{KNN}(q, k), q \in \mathcal{Q}_g\},$$
$$E_{\text{inter}} = \{(q, c_i) \mid q \in \mathcal{Q}_{g,i}^\alpha, c_i \in \mathcal{C}\},$$

where $E_{\text{intra}}$ encodes semantic similarity between queries, and $E_{\text{inter}}$ links queries to their source chunks. For simplicity, let $q$ denote $(q, a)$ and $q'$ denote $(q', a')$. In practice, $q$ corresponds to the combined query–answer pair $q \oplus a$ before being embedded into a vector space. Unless otherwise stated, we adopt this simplified notation throughout the following discussion. $\text{KNN}(q, k)$ denotes the set of the $k$ most similar queries $q'$ to $q$ based on the similarity function. The resulting *Query-Centric Graph* provides a high-quality, semantically rich, and granularity-controllable structure for downstream retrieval.

### 3.4 QUERY-CENTRIC GRAPH RETRIEVAL & GENERATION MECHANISM

This component performs retrieval and response generation over the constructed Query-Centric Graph. It consists of four sequential steps: (1) retrieving relevant queries, (2) expanding to neighboring queries, (3) collecting and ranking associated chunks, and (4) generating responses.

**Step 1: Retrieving Relevant Queries.** Given a user query $q_u$, we first compute its similarity with graph queries $\mathcal{Q}_g = \bigcup_{i=1}^{N} \mathcal{Q}_{g,i}^\alpha$. The similarity is defined as:

$$s(q_u, q) = \text{sim}(q_u, q) + \epsilon, \quad \forall q \in \mathcal{Q}_g,$$

where $\text{sim}(\cdot, \cdot)$ denotes cosine similarity, and $\epsilon = 1$. To construct the set of retrieved queries, we apply both a similarity threshold $\gamma \in [0, 2]$ to ensure relevance and a maximum node constraint $n$ to control graph size.

$$\mathcal{Q}_r = \{q \in \mathcal{Q}_g \mid s(q_u, q) \geq \gamma\}, \quad |\mathcal{Q}_r| \leq n.$$

**Step 2: Expanding to Neighboring Queries.** For each retrieved query $q \in \mathcal{Q}_r$, we collect its $h$-hop neighbors from the query-level graph. Formally, the one-hop neighbors are defined as: $\mathcal{H}^1(q) = \{q' \mid (q, q') \in E_{\text{intra}}\}$. For $h > 1$, the $h$-hop neighbors are obtained by expanding the $(h-1)$-hop set:

$$\mathcal{H}^n(q) = \bigcup_{q' \in \mathcal{H}^{n-1}(q)} \mathcal{H}^1(q').$$

The final query set is then the union of retrieved queries and their $h$-hop neighborhoods:

$$\mathcal{Q}^* = \mathcal{Q}_r \cup \bigcup_{q \in \mathcal{Q}_r} \bigcup_{i=1}^{h} \mathcal{H}^i(q).$$

**Step 3: Collecting and Ranking Associated Chunks.** Each query $q \in \mathcal{Q}^*$ is linked to its originating chunk(s) via query–chunk membership relations:

$$\mathcal{C}^* = \{c_i \in \mathcal{C} \mid (q, c_i) \in E_{\text{inter}}, \ q \in \mathcal{Q}^*\}.$$

To select the most relevant chunks, we compute a relevance score for each chunk $c \in \mathcal{C}^*$:

$$s(c) = \frac{1}{|\mathcal{Q}_c|} \sum_{q \in \mathcal{Q}_c} \text{sim}(q_u, \ q),$$

where $\mathcal{Q}_c$ denotes the subset of queries in $\mathcal{Q}^*$ associated with chunk $c$. Chunks are ranked by $s(c)$, and the top-$K$ are retained:

$$\mathcal{C}_{\text{top-}K} = \text{Top-}K\{c \in \mathcal{C}^* \mid s(c)\}.$$

**Step 4: Generating Responses.** Finally, the answer is generated by conditioning the LLM on the user query $q_u$ and the top-$K$ selected chunks $\mathcal{C}_{\text{top-}K}$:

$$a = \text{LLM}(q_u \mid \mathcal{C}_{\text{top-}K}).$$

The Query-Centric Graph Retrieval & Generation process enhances traditional RAG by retrieving through queries rather than directly over chunks, expanding to multi-hop neighbors, and aggregating evidence with controllable granularity. This design ensures better coverage of relevant document segments, improved response accuracy, and interpretable reasoning paths, distinguishing it from conventional chunk-based and graph-based RAG approaches.

## 4 EXPERIMENTS

### 4.1 EXPERIMENTAL SETTINGS

**Datasets.** We evaluate on two QA benchmarks: **LiHuaWorld** (Fan et al., 2025) and **MultiHop-RAG** (Tang & Yang, 2024). LiHuaWorld contains one year of English chat records from a virtual user, with queries spanning single-hop, multi-hop, and unanswerable types, each paired with annotated answers and supporting documents. MultiHop-RAG is constructed from English news articles and provides multi-hop queries with ground-truth answers and evidence. Together, these datasets cover long-term personal memory QA and news-based multi-hop reasoning. Detailed statistics are provided in Appendix A.

**Evaluation Metric.** We evaluate the RAG system outputs using automatic exact-match *Accuracy* and an *LLM-as-a-Judge* protocol (Gu et al., 2024). *Accuracy* is defined as the proportion of queries for which the predicted answer exactly matches the reference semantically. For *LLM-as-a-Judge*, we employ a strong instruction-tuned model (`Qwen2.5-72B-Instruct`) as the evaluator. The judge receives the user query, candidate response(s), and the reference answer, and outputs a categorical correctness decision (correct/incorrect). To ensure reproducibility, we adopt a fixed prompt and deterministic decoding.

**Baselines.** We compare QCG-RAG with representative RAG and graph-based RAG methods:

• **Naive RAG** (Lewis et al., 2020): A standard baseline that retrieves relevant documents via a dense retriever and conditions an LLM generator on the retrieved content.

• **D2QRAG & D2Q--RAG**: Extensions of Naive RAG that incorporate Doc2Query (Nogueira et al., 2019) or Doc2Query-- (Gospodinov et al., 2023) for document expansion. Doc2Query appends generated queries to documents, while Doc2Query-- further filters irrelevant or hallucinated queries.

| Method | Graph | Context | | | | LiHuaWorld | | | | MultiHop-RAG | | | | |
|---|---|---|---|---|---|---|---|---|---|---|---|---|---|---|
| | | C | E | R | S | Overall | Multi | Single | Null | Overall | Inference | Comparison | Temporal | Null |
| Naive RAG | - | ✓ | ✗ | ✗ | ✗ | 65.78% | 43.94% | 66.80% | **80.00%** | 75.80% | 93.46% | 66.85% | 68.00% | 71.21% |
| D2QRAG | - | ✓ | ✗ | ✗ | ✗ | 63.74% | 39.39% | 65.22% | 76.92% | 76.20% | 92.81% | 69.61% | 66.00% | 71.21% |
| D2Q--RAG | - | ✓ | ✗ | ✗ | ✗ | 64.99% | 40.91% | 66.60% | 76.92% | 76.80% | 92.16% | 67.96% | **70.00%** | 75.76% |
| GraphRAG | KG | ✓ | ✓ | ✓ | ✓ | 42.70% | 28.79% | 39.72% | **80.00%** | 67.20% | 81.05% | 64.09% | 66.00% | 45.45% |
| LightRAG | KG | ✗ | ✓ | ✓ | ✗ | 66.41% | 57.58% | 69.76% | 49.23% | 72.20% | **94.12%** | 71.82% | 59.00% | 42.42% |
| MiniRAG | KG | ✓ | ✓ | ✗ | ✗ | 60.28% | **62.12%** | 58.89% | 69.23% | 60.40% | 75.16% | 54.70% | 54.00% | 51.52% |
| KG-Retriever | HIG | ✗ | ✗ | ✓ | ✗ | 31.24% | 18.18% | 27.67% | 72.31% | 47.60% | 79.08% | 19.89% | 33.00% | 72.73% |
| QCG-RAG | QCG | ✓ | ✗ | ✗ | ✗ | **73.16%** | **62.12%** | **74.51%** | 73.85% | **79.60%** | 93.46% | **74.59%** | 69.00% | **77.27%** |

Table 1: Performance comparison on LiHuaWorld and MultiHop-RAG using Accuracy (%). Context indicates whether the method incorporates chunks (C), entities (E), relations (R), or summaries (S). Graph includes Knowledge Graph (KG), Hierarchical Index Graph (HIG), and Query-Centric Graph (QCG). The **best** results are highlighted in **bold**, and the second-best results are underlined.

• **GraphRAG** (Edge et al., 2024a): A graph-based approach that constructs entity-centric knowledge graphs and community-level summaries, and applies a map-reduce strategy over communities to aggregate answers. GraphRAG is implemented using MsGraphRAG-Neo4j[4], enabling retrieval over chunks, entities, relations, and summaries.

• **LightRAG** (Guo et al., 2024): A graph-based method that builds entity–relation indexes and employs a dual-level retrieval framework for fine-grained entity access and coarse-grained topic retrieval.

• **MiniRAG** (Fan et al., 2025): A lightweight heterogeneous graph framework that unifies text chunks and entities into a single index and employs heuristic retrieval for efficient knowledge discovery.

• **KG-Retriever** (Chen et al., 2024): A hierarchical index graph (HIG) framework combining an entity-level knowledge graph with a document-level layer to enhance both intra-document and cross-document connectivity.

**Setup Details.** Following prior work (Fan et al., 2025), we segment documents into chunks of 1200 tokens with an overlap of 100 tokens, and additionally use a 512-token with 64-token overlap setting for comparison. For embeddings, we employ Sentence-BERT (Reimers & Gurevych, 2019), using `all-MiniLM-L6-v2` as the default model and `all-mpnet-base-v2` for comparison; cosine similarity is used for retrieval without reranking. We retrieve the top-$K$ chunks ($K = 5$) and cap the input length at 6000 tokens.

For QCG-RAG, we set dataset-specific hyperparameters: for **LiHuaWorld**, $M = 20$, $\alpha = 80\%$, $h = 1$, $k = 2$, $n = 10$, and $\gamma = 1.5$; for **MultiHop-RAG**, $M = 20$, $\alpha = 80\%$, $h = 1$, $k = 3$, $n = 15$, and $\gamma = 1.0$. Details of the implementation can be found in Appendix B.

We adopt `Qwen2.5-72B-Instruct` (Team, 2024) for query generation to build query-centric graphs, and use `Qwen2.5-7B-Instruct` for comparison. We also employ `Qwen2.5-72B-Instruct` for question answering and response evaluation within the *LLM-as-a-Judge* framework. The LLM prompts are provided in Appendix C.

To assess the impact of node choice on QCG-RAG performance, we also conduct an ablation on node formulations using both LiHuaWorld and MultiHop-RAG. Specifically, we compare four variants: (1) *QCG w/ Query*, where only queries are used as nodes; (2) *QCG w/ Answer*, where only answers are used as nodes; (3) *Doc2Query w/ 7B*, where queries are generated by `Qwen2.5-7B-Instruct`; and (4) *Sentence Nodes*, where each sentence forms a graph node. The default setting of QCG-RAG employs concatenated *Query+Answer* nodes.

## 4.2 Experimental Results

**RQ1: How does QCG-RAG perform in QA compared with baselines?** Table 1 reports accuracy across LiHuaWorld and MultiHop-RAG. Overall, QCG-RAG achieves the best performance among all baselines, with an average accuracy of 73.16% on LiHuaWorld and 79.60% on MultiHop-RAG. Compared to Naive RAG (65.78% / 75.80%), QCG-RAG yields consistent gains of +7.4 and +3.8

---

[4] https://github.com/neo4j-contrib/ms-graphrag-neo4j

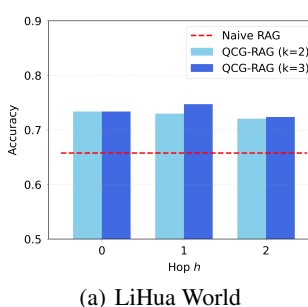 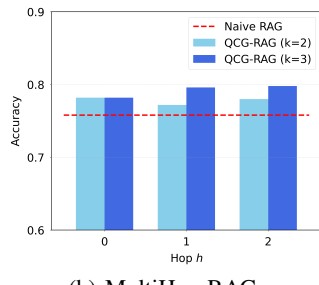

| (a) LiHua World | (b) MultiHop-RAG |

Figure 3: QA performance with zero-hop, one-hop, and two-hop retrieval over Query-Centric Graphs (QCG) on (a) LiHuaWorld and (b) MultiHop-RAG. Naive RAG (red dashed line) is shown as the baseline, while QCG-RAG results are reported for $k = 2$ and $k = 3$.

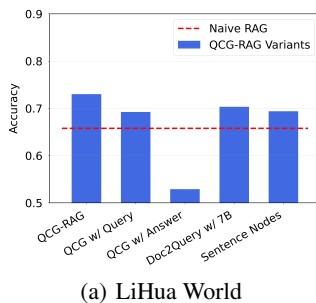 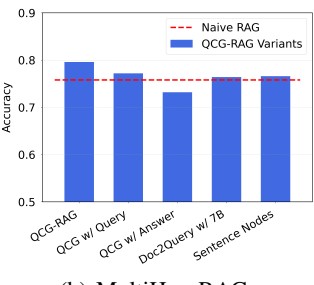

| (a) LiHua World | (b) MultiHop-RAG |

Figure 4: Ablation on node choices in QCG-RAG across (a) LiHuaWorld and (b) MultiHop-RAG.

points, demonstrating that query-centric indexing substantially improves retrieval quality and answer correctness. On the more challenging multi-hop subsets, QCG-RAG notably outperforms prior graph-based methods: 62.12% vs. 57.58% (LightRAG) on LiHuaWorld multi-hop, and 74.59% vs. 71.82% (LightRAG) on MultiHop-RAG comparison queries. Importantly, QCG-RAG maintains competitive performance across null queries (73.85% / 77.27%), showing robustness against unanswerable cases where many graph-based baselines degrade sharply. These results confirm that balancing graph granularity via query-centric construction yields stronger semantic alignment, leading to superior QA accuracy across both long-term personal memory and news-based multi-hop reasoning tasks.

**RQ2: Why does QCG-RAG improve QA performance for multi-hop questions?** Figure 3 shows the effect of hop size $h$ on QA accuracy across LiHuaWorld and MultiHop-RAG. QCG-RAG consistently outperforms Naive RAG under all hop settings, with the largest gains observed for one-hop and two-hop settings. For instance, on MultiHop-RAG, QCG-RAG ($k=3$) improves accuracy from 75.8% (Naive RAG) to nearly 80%, demonstrating stronger multi-hop reasoning. This advantage arises because query-centric graphs explicitly encode query-to-query and query-to-chunk relations, enabling effective expansion from initial retrieved queries to semantically related neighbors. By leveraging these structured connections, QCG-RAG can capture dispersed evidence across documents and integrate it into coherent reasoning chains, thereby addressing the limitations of chunk-only retrieval that often misses intermediate links. Case studies on both datasets are provided in Appendix D to illustrate scenarios where QCG-RAG succeeds in deriving correct answers through multi-hop retrieval over QCG, whereas Naive RAG fails.

**RQ3: What are the effects of QCG nodes in QCG-RAG?** Figure 4 presents an ablation study on different node choices in QCG-RAG. The full model, which uses concatenated *Query+Answer* as nodes, consistently achieves the best accuracy on both LiHuaWorld and MultiHop-RAG. Using only queries (*QCG w/ Query*) yields competitive but lower performance, while using only answers (*QCG w/ Answer*) results in a substantial drop, highlighting that queries provide essential semantic grounding while answers alone lack sufficient context. Moreover, replacing the 72B-based Doc2Query with a smaller generator (*Doc2Query w/ 7B*) leads to reduced performance, indicating that graph quality benefits from stronger generators but ultimately depends more on node formulation than model size alone. Finally, constructing graphs with sentence-level nodes (*Sentence Nodes*) underperforms QCG,

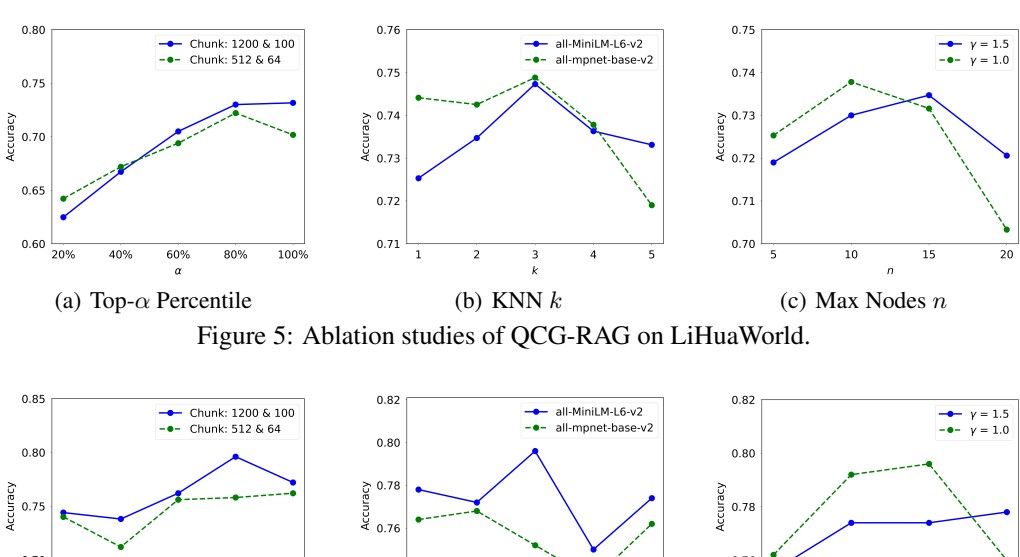

Figure 5: Ablation studies of QCG-RAG on LiHuaWorld.

Figure 6: Ablation studies of QCG-RAG on MultiHop-RAG.

confirming that query-centric granularity provides a better balance between semantic richness and interpretability. Overall, these results demonstrate that QCG nodes, particularly the *Query+Answer* concatenation, are critical to achieving robust retrieval and reasoning performance.

**RQ4: How do the hyperparameters, such as $\alpha$, $k$ and $n$, affect the performance of QCG-RAG?** Figures 5 and 6 present ablations of key hyperparameters in QCG-RAG: (a) the effect of query truncation at the top-$\alpha$ percentile under different chunking strategies; (b) the effect of the number of nearest neighbors per node $k$ under different embedding models; and (c) the effect of the maximum number of retrieved query nodes $n$ under different similarity thresholds $\gamma$. First, the query truncation top-$\alpha$ percentile (Figures 5a, 6a) shows a clear upward trend from 20% to 80%, after which performance plateaus or slightly decreases, suggesting that moderate query pruning effectively balances noise reduction and coverage. Second, the number of the nearest neighbors per node $k$ (Figures 5b, 6b) peaks around $k = 3$, reflecting that too few neighbors restrict multi-hop associations on QCG, whereas too many introduce noisy or redundant connections. Finally, the maximum number of retrieved nodes $n$ (Figures 5c, 6c) achieves optimal performance when set between 10 and 15. A smaller $n$ (e.g., 5) limits retrieval coverage, while a larger $n$ (e.g., 20) introduces noisy queries that degrade accuracy. Overall, the findings underscore that carefully balanced hyperparameters (e.g., $\alpha$=80%, $k$=3, $n$=15) are crucial for QCG-RAG, as they enable adequate evidence coverage while avoiding redundancy and noise from excessive expansion.

## 5 CONCLUSION

We presented QCG-RAG, a query-centric graph retrieval-augmented generation framework that addresses the granularity dilemma in existing graph-based RAG methods. Unlike prior approaches that rely on either coarse document-level graphs or fine-grained entity-centric graphs, QCG-RAG constructs controllable-granularity query-centric graphs by leveraging Doc2Query and Doc2Query- -, and employs a tailored multi-hop retrieval process over these graphs. Extensive experiments on LiHuaWorld and MultiHop-RAG demonstrate that QCG-RAG consistently outperforms both chunk-based and graph-based baselines, achieving state-of-the-art performance on multi-hop and long-context question answering. Our ablations further show that query formulation, node design, and balanced hyperparameter settings are critical to performance.

## LIMITATIONS

While QCG-RAG advances the state of graph-based retrieval-augmented generation, it still has several limitations. First, the framework depends on query generation quality; errors or biases in Doc2Query may propagate into graph construction and retrieval. Second, although query-centric graphs mitigate token overhead compared to entity-level graphs, constructing and maintaining large-scale QCGs remains computationally costly when applied to web-scale corpora. Third, our experiments are limited to English QA benchmarks; extending QCG-RAG to multi-lingual or domain-specific scenarios (e.g., legal or biomedical text) requires further validation. Finally, the current retrieval mechanism primarily focuses on structural and semantic similarity, but does not incorporate advanced reasoning strategies such as reinforcement learning or self-reflection, which may further enhance complex reasoning. We leave these directions for future work.

## ETHICS STATEMENT

This work builds on QA benchmarks (LiHuaWorld and MultiHop-RAG) that contain either synthetic or publicly available text, without involving private or sensitive user data. The proposed QCG-RAG framework aims to improve retrieval-augmented reasoning and does not introduce additional risks beyond those inherent to large language models, such as potential hallucinations or biases inherited from pre-trained models. We caution against deploying QCG-RAG in high-stakes domains (e.g., healthcare, law) without rigorous domain-specific validation, and emphasize that our contributions should be viewed as methodological advances in retrieval and reasoning.

## REPRODUCIBILITY STATEMENT

We have provided detailed descriptions of the QCG-RAG framework, including graph construction, retrieval mechanisms, and hyperparameter choices, in the main text and appendix. Dataset statistics, evaluation metrics, and experimental setups (embedding models, LLM configurations, and query generation methods) are reported in full. Hyperparameter ranges and default settings for both LiHuaWorld and MultiHop-RAG are explicitly specified, and all prompts used for query generation and evaluation are included in the appendix. Together, these details ensure that our experiments can be reliably reproduced.

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

## A    DATASET STATISTICS

**LiHuaWorld.**   LiHuaWorld is an English dialogue dataset comprising one year of conversational records from a virtual user, spanning diverse daily-life topics such as social interactions, fitness, entertainment, and personal affairs. It contains **442 documents** and **637 queries**, including **506 single-hop**, **66 multi-hop**, and **65 unanswerable (null)** questions. Each query is paired with manually annotated answers and supporting documents, allowing fine-grained evaluation of both retrieval and reasoning.

**MultiHop-RAG.**   MultiHop-RAG is a news-based multi-hop QA dataset constructed from English news articles. It consists of **609 documents** and **2,556 queries** with ground-truth answers and supporting evidence. We use the first **500 queries** as the test set, primarily to ensure computational feasibility and enable extensive ablation studies, while maintaining coverage across query types. Queries are categorized into **Inference (153)**, **Comparison (181)**, **Temporal (100)**, and **Null (66)** types.

Together, these datasets provide complementary evaluation scenarios: LiHuaWorld targets long-term personal memory QA, while MultiHop-RAG emphasizes news-based multi-hop reasoning.

## B    IMPLEMENTATION DETAILS

### B.1    EMBEDDING MODEL SETTINGS

We employed pre-trained sentence embedding models from the `Sentence-Transformers`[5] library for encoding text segments.  Unless otherwise specified, the default model is `all-MiniLM-L6-v2`, which produces 384-dimensional embeddings. This model is lightweight and optimized for efficiency, making it suitable for large-scale retrieval scenarios. For comparison, we also experimented with `all-mpnet-base-v2`, a stronger encoder that outputs 768-dimensional embeddings. While this model provides higher representational capacity and improved semantic alignment, it incurs increased computational and memory costs. Both models were applied in a zero-shot manner without task-specific fine-tuning. During retrieval, embeddings were $\ell_2$-normalized before similarity computation to ensure stable and consistent cosine similarity scores.

### B.2    LLM SETTINGS

All large language models (LLMs) were accessed via the BaiLian[6] API platform with default generation parameters: $temperature = 0.7$, $top\_p = 0.8$, $top\_k = 20$, $max\_input\_tokens = 129{,}024$, and $max\_tokens = 8{,}192$. For query generation, question answering, and evaluation, we primarily employed `Qwen2.5-72B-Instruct`. For comparison, we also considered `Qwen2.5-3B-Instruct` in query-centric graph construction. The 72B model was mainly applied to computationally intensive tasks requiring stronger reasoning capacity and long-context processing. Unless otherwise specified, all models were used in a zero-shot setting without task-specific fine-tuning.

### B.3    QCG-RAG HYPERPARAMETERS

We report the default hyperparameter settings for each dataset and the candidate search ranges considered during tuning.

---

[5]`https://huggingface.co/sentence-transformers`
[6]`https://bailian.console.aliyun.com/`

**Default Values.** For the LiHuaWorld dataset, we set $M = 20$, $\alpha = 80\%$, $h = 1$, $k = 2$, $n = 10$, and $\gamma = 1.5$. For the MultiHop-RAG dataset, we set $M = 20$, $\alpha = 80\%$, $h = 1$, $k = 3$, $n = 15$, and $\gamma = 1.0$.

**Candidate Ranges.** The hyperparameters were selected from the following ranges: $\alpha \in \{20\%, 40\%, 60\%, 80\%, 100\%\}$, $k \in \{1, \ldots, 5\}$, $h \in \{0, 1, 2\}$, $n \in \{5, 10, 15, 20\}$, and $\gamma \in \{1.0, 1.5\}$. Here, $M$ is the number of generated queries per chunk, $\alpha$ controls the query truncation ratio, $h$ denotes the number of retrieval hops, $k$ specifies the number of retrieved query nodes, $n$ is the number of neighbors per node, and $\gamma$ is the similarity threshold. Unless otherwise specified, reported results correspond to the default settings.

## C LLM PROMPTS

We show LLM prompts for query generation and answer evaluation, including Doc2Query Prompt, Response Generation Prompt, and Evaluation Prompt.

### C.1 DOC2QUERY PROMPT

To generate diverse supervision signals for query-centric graph construction, we adopted a standardized `Doc2Query` prompt. This prompt instructs the model to produce multiple distinct queries from each text chunk, along with exact answers grounded in the chunk. The full prompt is shown in Prompt C.1.

---

**Prompt C.1: Doc2Query Prompt**

```
---Role---

You are a **Doc2Query** assistant.

---Goal---

Given a text chunk, generate 20 distinct user queries that can be
    directly answered by that chunk.
For each query, also provide an exact answer and a relevance score.

---Generation rules---

1. **Answerability** - Every query must be answerable using only
    information in the chunk.

2. **Comprehensive coverage** - Collectively, the all generated
    queries should cover all key ideas in the chunk from different
    viewpoints or levels of detail.

3. **Diversity requirements** - Ensure diversity along the following
    dimensions:
    - *Question-style variety* - Mix interrogative forms (who/what/why/
        how/where/when/did), imperative prompts ("List...", "Summarize
        ..."), comparative questions, conditional or speculative forms,
        etc.
    - *Content-perspective variety* - Include queries on facts,
        definitions, methods, reasons, outcomes, examples, comparisons,
        limitations, and so on.
    - *Information granularity* - Combine macro (overall purpose, high-
        level summary) and micro (specific figures, terms, steps)
        queries.
    - *User-intent variety* - Simulate intents such as confirmation,
        evaluation, usability, diagnosis, and decision-making (e.g., "Is
        this approach more efficient than ...?").
```

```
    - *Linguistic expression variety* - Vary wording, syntax (active
        <--> passive), and synonyms; avoid repeating near-identical
        phrases.
    - *No redundancy* - Each query must be meaningfully distinct;
        eliminate trivial rephrases that offer no new angle.
    - *Chunk-grounded specificity* - Queries must be grounded in
        specific factual points from the chunk. Avoid vague or generic
        formulations such as "What did X say?" or "Tell me more about Y"
         that lack anchoring in actual content.

4. **Required fields** - Each output item must be based on the given
    chunk and include the following fields:
    - **query** - A question or search phrase a user might ask.
    - **answer** - A concise answer taken verbatim (or nearly verbatim
        ) from the chunk.

---Example---

1. **Input Chunk**
    - Alice met with Bob at the Central Cafe on Tuesday to discuss
        their upcoming collaborative research project. During the
        meeting, Bob suggested incorporating advanced AI methodologies
        into their experimental design, which Alice enthusiastically
        supported. They agreed to present their initial findings at the
         International AI Conference next March.

2. **Generated Queries**
    - Where did Alice and Bob meet?
    - When did the meeting take place?
    - What was the main topic discussed during the meeting?
    - Who suggested incorporating advanced AI methodologies?
    - Did Alice support Bob's suggestion about AI methodologies?
    - When will Alice and Bob present their initial findings?
    - At which conference will their findings be presented?
    - What kind of methodologies were discussed?
    - What is the nature of Alice and Bob's project?
    - Who participated in the meeting at Central Cafe?

---Output format---

Return *only* the following JSON array; do not include any additional
    text. Include an `index` field for each query.

[
  {"index": 0, "query": "", "answer": ""},
  {"index": 1, "query": "", "answer": ""},
  ...
  {"index": 19, "query": "", "answer": ""}
]
```

## C.2 RESPONSE GENERATION PROMPT

To guide model responses grounded in tabular data, we employed a standardized prompt that specifies the assistant's role, objective, and the required response length and format. The full prompt is shown in Prompt C.2.

**Prompt C.2: Response Generation Prompt**

```
---Role---
```

```
You are a helpful assistant responding to questions about data in the
    tables provided.

---Goal---

Generate a response of the target length and format that responds to
    the user's question, summarizing all information in the input data
    tables appropriate for the response length and format, and
    incorporating any relevant general knowledge.
If you don't know the answer, just say so. Do not make anything up.
Do not include information where the supporting evidence for it is not
     provided.

---Target response length and format---

Multiple Paragraphs

Add sections and commentary to the response as appropriate for the
    length and format. Style the response in markdown.
```

## C.3 EVALUATION PROMPT

To ensure consistent and transparent evaluation of candidate answers generated by retrieval-augmented generation (RAG), we employed a standardized prompt for judgment. The prompt specifies the evaluation role, criteria, and output format, focusing on factual accuracy and completeness. The full evaluation prompt is shown in Prompt C.3.

**Prompt C.3: Evaluation Prompt**

```
---Role---

You are a helpful evaluation assistant.
You will be given a question, a gold-standard answer, and a candidate
    answer generated via retrieval-augmented generation (RAG).

---Goal---

Evaluate the candidate answer against the gold-standard answer based
    on factual accuracy and completeness in answering the question.

Scoring Criteria:
- score=1 (Correct): The candidate answer is factually accurate and
    fully or reasonably paraphrases the gold-standard answer.
- score=0 (Incorrect): The candidate answer is factually incorrect,
    irrelevant, incomplete, or does not answer the question.

---Output Format---

Provide your evaluation in the following JSON format:

{
  "score": X
}
where X is either 1 or 0.
```

## D    CASE STUDIES

### D.1    A CASE STUDY ON LIHUAWORLD

We present an example of question answering with supporting evidence chunks on LiHuaWorld in Example D.1. From the example, it can be observed that the supporting evidence chunks "20261219_19:00-0", "20261220_20:00-0", and "20261228_10:00-0" provide only one part of the answer, namely "LiHua," whereas "20261221_12:00-0" contains all answer entities: "LiHua," "Chae Song-hwa," and "Yuriko Yamamoto."

---

**Example D.1: An example of question answering on LiHuaWorld**

```
{
"question": "Who knows about Wolfgang going to Hong Kong?",
"answer": "LiHua & Chae & Yuriko",
"type": "Multi",
"evidence":
[
{"doc_id": "20261219_19:00",
"chunk_id": "20261219_19:00-0",
"chunk": "WolfgangSchulz: Hey! Just a heads up, I'm off to Hong Kong
    for a couple of days next week. Anything you want me to grab while
    I'm there?\nLiHua: Ooh, nice trip! Maybe look for some cool snacks
    or a local souvenir? That would be awesome!\nWolfgangSchulz: Sounds
     good! I'll keep an eye out for some unique snacks and a souvenir.
    Anything specific you had in mind?\nLiHua: Not really, just
    something that screams Hong Kong! I'm sure whatever you find will
    be great!\nWolfgangSchulz: Okay, I got it! I'll make sure to find
    something special for you. \nLiHua: Thanks, man! Have fun on your
    trip! Safe travels!\nWolfgangSchulz: Appreciate it! I'll keep you
    updated. \n"},
{"doc_id": "20261220_20:00",
"chunk_id": "20261220_20:00-0",
"chunk": "WolfgangSchulz: Hey! I've been looking into some techniques
    for taking stunning photos. Got any tips? I'm heading to Hong Kong
    soon and want to capture some beautiful shots! \nLiHua: Oh, that
    sounds awesome! For taking great photos, try experimenting with the
     golden hour for the best lighting-sunrise or sunset can work
    wonders! Also, don't forget to compose your shots with leading
    lines to draw the viewer's eye. Have fun and shoot a ton!\
    nWolfgangSchulz: Thanks for the tips! I'll definitely aim for the
    golden hour. Anything else I should keep in mind while I'm shooting
    ? \nLiHua: Make sure to play with different angles! Getting low or
    high can really change the mood of your shots. Also, don't hesitate
     to include some locals or unique elements in your frame-it can add
     a lot of life to your photos! Enjoy the adventure!\nWolfgangSchulz
    : Great advice! I'll try to capture the vibe of the place and the
    locals. Can't wait to show you the photos! \nLiHua: I can't wait to
     see them! You'll have an amazing time in Hong Kong. Let me know if
     you need more tips! "},
{"doc_id": "20261221_12:00",
"chunk_id": "20261221_12:00-0",
"chunk": "WolfgangSchulz: Hey everyone! Just a heads-up, I won't be
    able to make it to practice this week. I have a business trip to
    Hong Kong. Catch you all next time!\nWolfgangSchulz: Hope you all
    have a great jam without me! Looking forward to hearing what you
    come up with. \nYurikoYamamoto: Aww, Wolfgang, we'll miss you! But
    safe travels! We'll make sure to save some cool tunes for when you'
    re back! \nWolfgangSchulz: Thanks! I appreciate it. I'll be eager
    to catch up once I'm back and hear all the new stuff! \
```

---

```
      nWolfgangSchulz: Li Hua, make sure you all have enough fun for me!
      Got any songs lined up for this week?\nYurikoYamamoto: Let's see
      how it goes! Maybe we can try something new. How about \"Chasing
      Cars\" or \"Tears in Heaven\"? What do you think?\nChaeSong-hwa:
      Sounds great! Those songs would be awesome to try out. Have fun!
      Can't wait to hear how it goes!\nWolfgangSchulz: Can't wait to hear
       it too! Keep the vibes going while I'm away. Let's save those
      tunes for my return! \nLiHua: Btw, do you guys want to try some new
       harmonies in \"Chasing Cars\"? I think it would sound amazing!\n
      YurikoYamamoto: That sounds like a fantastic idea! Harmonies always
       add such a nice touch. Let's give it a shot! \nLiHua: Chae, you
      think we can also add some cool rhythms? I'm ready to get creative
      !\nChaeSong-hwa: Absolutely! I love the idea of mixing in some cool
       rhythms. It'll make our sound even more exciting! Can't wait to
      see how it turns out!\nYurikoYamamoto: Yes! It's going to be so fun
      . I'm looking forward to our practice this week! \nChaeSong-hwa:
      Just a reminder to share voice notes or recordings so Wolfgang can
      join in remotely! Would love to keep him in the loop!\nLiHua: Great
       idea! I'll make sure to take some recordings. Wolfgang, you won't
      miss out on any jams! \nWolfgangSchulz: Thanks, everyone! You guys
      have fun and I'll catch up once I'm back. Don't forget to send me
      the recordings! I'm excited to hear everything! You all have fun
      jamming! \n"}
 {"doc_id": "20261228_10:00",
 "chunk_id": "20261228_10:00-0",
 "chunk": "WolfgangSchulz: Hey! Do you have some time after work today?
      I want to share some awesome pics I took in Hong Kong! \nLiHua:
      Hey! That sounds amazing! I'd love to see those pictures. What time
       were you thinking?\nWolfgangSchulz: How about 7 pm? We can grab
      some food and I can show you the pics then! \nLiHua: 7 pm works for
       me! Can't wait to see those stunning shots! What food are you in
      the mood for?\nWolfgangSchulz: I'm thinking maybe some Thai food?
      It's been a while since we had that! What do you think? \nLiHua:
      Thai food sounds perfect! I'm in! See you at 7! \n"}
 ]
 }
```

**Naive RAG on LiHuaWorld.**  We provide an illustrative example of a query–answer evaluation case produced by Naive RAG. The instance is shown in Use Case D.1. From the use case, it can be observed that Naive RAG, in its Top-5 chunk ranking, retrieved only "20261219_19:00-0" and "20261228_10:00-0," which record conversations between Wolfgang Schulz and Li Hua about the Hong Kong trip, and thus evidently produced an incorrect answer.

---

**Use Case D.1: Naive RAG on LiHuaWorld**

```
{
"question": "Who knows about Wolfgang going to Hong Kong?",
"answer": "LiHua & Chae & Yuriko",
"evidence": ["20261219_19:00", "20261220_20:00", "20261221_12:00", "
    20261228_10:00"],
"type": "Multi",
"results":
{
"topk_chunk_ids": ["20261219_19:00-0", "20261223_23:00-0", "20260307
    _15:00-0", "20261228_10:00-0", "20260801_19:00-0"],
"response": "Wolfgang Schulz informed Li Hua about his trip to Hong
    Kong. Here's a summary of their conversation:

    ### Initial Announcement
```

---

```
      - **Wolfgang Schulz:** Announced he was going to Hong Kong for a
         few days and asked if Li Hua wanted him to bring back any
         snacks or souvenirs.
      - **Li Hua:** Suggested looking for cool snacks or a local
         souvenir, emphasizing something that 'screams Hong Kong.'

      ### Arrival and Plans
      - **Wolfgang Schulz:** Updated Li Hua upon arrival, expressing
         excitement about the vibrant city and mentioning plans to visit
          Victoria Peak and try dim sum.
      - **Li Hua:** Recommended trying street food in Mong Kok,
         particularly egg waffles, and asked Wolfgang to take photos.

      ### Sharing Experiences
      - **Wolfgang Schulz:** Later, invited Li Hua to see the photos
         from his trip, suggesting they meet for dinner at 7 pm.
      - **Li Hua:** Agreed to the plan, showing enthusiasm for seeing
         the photos and suggested Thai food for dinner.

      From these interactions, it is clear that Li Hua is aware of
         Wolfgang's trip to Hong Kong and has been actively engaged in
         discussing his experiences and plans."
  },
  "score": 0
  }
```

**QCG-RAG on LiHuaWorld.** We also provide an illustrative example of a query–answer evaluation case produced by our proposed QCG-RAG. The instance is shown in Use Case D.2. In addition, the generated Query+Answer text information corresponding to topk_qids and extra_qids_via_knn is presented in the Detail section. The q_id consists of a document ID $x$, a chunk ID $y$, and a generation index $z$, in the format "$x$-$y$-$z$" (by default, 20 queries are generated, so $z \in [0, 19]$). For example, "20261221_12:00-0-1" denotes the chunk "20261221_12:00", with "0" as the chunk ID and "1" as the generation index.

From the use case, it can be seen that QCG-RAG, by directly retrieving generated Query+Answer pairs, can accurately match those produced by the "20261219_19:00-0," and "20261228_10:00-0" chunks, as well as the query associated with the chunk "20261221_12:00-0" that contains all correct answers:

```
    {"q_id": "20261221_12:00-0-14",
     "query": "What does Yuriko Yamamoto
         say about Wolfgang's trip?",
      "answer": "Aww, Wolfgang, we'll miss
         you! But safe travels!"}
```

Furthermore, through one-hop expansion on the query-centric graph (QCG), it can additionally retrieve another query from the same chunk:

```
    {"q_id": "20261221_12:00-0-12",
     "query": "What does Wolfgang say about
         catching up after his trip?",
      "answer": "I'll be eager to catch up
         once I'm back and hear all the new
         stuff!"}
```

This enables the relevance score of "20261221_12:00-0" to be further enhanced in subsequent chunk ranking.

### Use Case D.2: QCG-RAG on LiHuaWorld

```
{
"question": "Who knows about Wolfgang going to Hong Kong?",
"answer": "LiHua & Chae & Yuriko",
"evidence": ["20261219_19:00", "20261220_20:00", "20261221_12:00", "
    20261228_10:00"],
"type": "Multi",
"results":
{
"topk_qids": ["20261221_12:00-0-1", "20261219_19:00-0-0", "20261219_19
    :00-0-8", "20261219_19:00-0-1", "20261223_23:00-0-0", "20261223_23
    :00-0-12", "20261219_19:00-0-15", "20261219_19:00-0-2", "20261223
    _23:00-0-1", "20261223_23:00-0-9", "20261228_10:00-0-3", "20260106
    _20:00-0-7", "20261228_10:00-0-0", "20261221_12:00-0-14", "20261228
    _10:00-0-1"],
"extra_qids_via_knn": ["20260428_18:00-0-5", "20260627_12:00-0-0",
    "20260307_15:00-0-9", "20261221_12:00-0-12"]
"chunk_to_queries": {"20260106_20:00-0": ["20260106_20:00-0-7"],
    "20260428_18:00-0": ["20260428_18:00-0-5"], "20260627_12:00-0":
    ["20260627_12:00-0-0"], "20261219_19:00-0": ["20261219_19:00-0-2",
    "20261219_19:00-0-1", "20261219_19:00-0-0", "20261219_19:00-0-15",
    "20261219_19:00-0-8"], "20261228_10:00-0": ["20261228_10:00-0-1", "
    20261228_10:00-0-3", "20261228_10:00-0-0"], "20260307_15:00-0":
    ["20260307_15:00-0-9"], "20261221_12:00-0": ["20261221_12:00-0-1",
    "20261221_12:00-0-14", "20261221_12:00-0-12"], "20261223_23:00-0":
    ["20261223_23:00-0-0", "20261223_23:00-0-12", "20261223_23:00-0-1",
    "20261223_23:00-0-9"]},
"sorted_chunk_ids": ["20261219_19:00-0", "20261223_23:00-0", "20261221
    _12:00-0", "20260106_20:00-0", "20261228_10:00-0", "20260307_15
    :00-0", "20260627_12:00-0", "20260428_18:00-0"],
"topk_chunk_ids": ["20261219_19:00-0", "20261223_23:00-0", "20261221
    _12:00-0", "20260106_20:00-0", "20261228_10:00-0"],
"response": "Wolfgang's trip to Hong Kong was known by several people,
    as evidenced by the conversations in the provided chats:

    ### Colleagues and Friends
    - **Li Hua**: Li Hua was one of the first to know about Wolfgang's
        trip. She suggested he bring back some cool snacks or a local
        souvenir from Hong Kong. She also gave him recommendations for
        places to visit and foods to try, such as Victoria Peak, Mong
        Kok street food, and egg waffles.
    - **Yuriko Yamamoto**: Yuriko knew about Wolfgang's business trip
        and expressed that the band would miss him during practice. She
        assured him they would save some cool tunes for when he
        returned and suggested songs like \"Chasing Cars\" and \"Tears
        in Heaven\" for the practice session.
    - **Chae Song-hwa**: Chae was also aware of Wolfgang's absence due
        to his trip. She agreed to try new harmonies and rhythms in \"
        Chasing Cars\" and reminded the group to share voice notes or
        recordings so Wolfgang could stay in the loop.

    ### Summary
    Wolfgang informed his friends and bandmates about his trip to Hong
        Kong. Li Hua, Yuriko Yamamoto, and Chae Song-hwa were all
        aware of his travel plans and discussed various aspects of his
        trip, including bringing back souvenirs, exploring the city,
        and keeping him updated on band activities."
},
"score": 1
}
```

**Detail D.1: Details of `qids` with QCG-RAG on LiHuaWorld**

```
{
"topk_qids":
[
{"q_id": "20261221_12:00-0-1", "query": "Who is Wolfgang going to
    visit in Hong Kong?", "answer": "Not specified, just a business
    trip."},
{"q_id": "20261219_19:00-0-0", "query": "Where is WolfgangSchulz going
    ?", "answer": "Hong Kong"},
{"q_id": "20261219_19:00-0-8", "query": "Who is traveling to Hong Kong
    ?", "answer": "WolfgangSchulz"},
{"q_id": "20261219_19:00-0-1", "query": "How long will WolfgangSchulz
    be in Hong Kong?", "answer": "a couple of days"},
{"q_id": "20261223_23:00-0-0", "query": "Where did WolfgangSchulz just
     arrive?", "answer": "Hong Kong"},
{"q_id": "20261223_23:00-0-12", "query": "What does LiHua think
    WolfgangSchulz will feel about Hong Kong?", "answer": "going to
    love it"},
{"q_id": "20261219_19:00-0-15", "query": "What did WolfgangSchulz say
    about his travel plans?", "answer": "I'm off to Hong Kong for a
    couple of days next week"},
{"q_id": "20261219_19:00-0-2", "query": "When is WolfgangSchulz
    leaving for Hong Kong?", "answer": "next week"},
{"q_id": "20261223_23:00-0-1", "query": "How does WolfgangSchulz
    describe Hong Kong?", "answer": "incredible, so vibrant and full of
     life"},
{"q_id": "20261223_23:00-0-9", "query": "What is WolfgangSchulz
    excited about in Hong Kong?", "answer": "explore, taste everything
    "},
{"q_id": "20261228_10:00-0-3", "query": "Where did WolfgangSchulz take
     the pictures?", "answer": "Hong Kong"},
{"q_id": "20260106_20:00-0-7", "query": "How does Wolfgang feel about
    exploring the city with LiHua?", "answer": "It'll be fun to catch
    up and explore together."},
{"q_id": "20261228_10:00-0-0", "query": "What does WolfgangSchulz want
     to share?", "answer": "some awesome pics I took in Hong Kong"},
{"q_id": "20261221_12:00-0-14", "query": "What does Yuriko Yamamoto
    say about Wolfgang's trip?", "answer": "Aww, Wolfgang, we'll miss
    you! But safe travels!"},
{"q_id": "20261228_10:00-0-1", "query": "Who is WolfgangSchulz
    planning to meet after work?", "answer": "LiHua"}
],
"extra_qids_via_knn":
[
{"q_id": "20260428_18:00-0-5", "query": "Who is WolfgangSchulz
    inviting to dinner?", "answer": "LiHua"},
{"q_id": "20260627_12:00-0-0", "query": "Who invited LiHua to hang out
     after work?", "answer": "WolfgangSchulz"},
{"q_id": "20260307_15:00-0-9", "query": "What does WolfgangSchulz say
    about seeing LiHua?", "answer": "Looking forward to it. It's been a
     while since we hung out."},
{"q_id": "20261221_12:00-0-12", "query": "What does Wolfgang say about
    catching up after his trip?", "answer": "I'll be eager to catch up
    once I'm back and hear all the new stuff!"}
]
}
```

## D.2 A CASE STUDY ON MULTIHOP-RAG

We present an example of question answering with supporting evidence chunks on MultiHop-RAG in Example D.2. The example question is of the type "comparison query," which requires leveraging

two long documents, "doc-451" and "doc-167," to answer. Each document can be segmented into multiple chunks, with each chunk containing no more than 1200 tokens.

---

**Example D.2: An example of question answering on Multihop-RAG**

```
{
"question": "Does 'The Age' article suggest that Australia's Davis Cup
    team is aiming for an improvement in their performance compared to
    the previous year, while the 'Sporting News' article indicates
    that the South Africa national rugby team has already achieved an
    improvement to reach the Rugby World Cup semi-finals?",
"answer": "Yes",
"evidence":
[
{"doc_id": "doc-451",
"chunk_id": ["doc-451-chunk-0", "doc-451-chunk-1", "doc-451-chunk-2"],
"title": ""Biggest win of my career": De Minaur, Popyrin power
    Australia into Davis Cup final",
"author": "Ian Chadband",
"url": "https://www.theage.com.au/sport/tennis/biggest-win-of-my-
    career-de-minaur-popyrin-power-australia-into-davis-cup-final
    -20231125-p5emr5.html?ref=rss&utm_medium=rss&utm_source=rss_sport",
"source": "The Age",
"category": "sports",
"published_at": "2023-11-24T23:10:22+00:00",
"fact": ""Hopefully we can go one better this year," he added,
    recalling the 2-0 defeat by Canada in 2022."
},
{"doc_id": "doc-167",
"chunk_id": ["doc-167-chunk-0", "doc-167-chunk-1"],
"title": "Where to watch England vs South Africa: Live stream, TV
    channel, lineups, odds for 2023 Rugby World Cup semifinal",
"author": "Jonathan Burnett",
"url": "https://www.sportingnews.com/us/rugby-union/news/watch-england
    -vs-south-africa-stream-channel-rugby-world-cup/20
    ce7b361138e7c1a1574572",
"source": "Sporting News",
"category": "sports", "published_at": "2023-10-20T22:00:05+00:00", "
    fact": "England will no doubt be looking to echo the spirit of
    2019, when they beat the All Blacks 19-7 to reach the final, but in
     South Africa, they face a side who have taken their game to new
    heights just to reach the semis."
}
],
"type": "comparison_query"
}
```

---

**Naive RAG on MultiHop-RAG.** We provide an illustrative example of a query–answer evaluation case produced by Naive RAG. The instance is shown in Use Case D.3. From the experimental results, Naive RAG failed to retrieve the correct article segment "doc-451" using the keywords ``The Age'' article and Australia's Davis Cup team. Instead, it only captured "doc-167-chunk-0," which led to a misinterpretation during the comparison process: the target entity "Australia's Davis Cup Team" was mistakenly understood as "Australian Rugby (Wallabies)."

---

**Use Case D.3: Naive RAG on MultiHop-RAG**

```
{
"question": "Does 'The Age' article suggest that Australia's Davis Cup
    team is aiming for an improvement in their performance compared to
    the previous year, while the 'Sporting News' article indicates
```

---

```
     that the South Africa national rugby team has already achieved an
     improvement to reach the Rugby World Cup semi-finals?",
"answer": "Yes",
"evidence":
[
{"doc_id": "doc-451",
"chunk_id": ["doc-451-chunk-0", "doc-451-chunk-1", "doc-451-chunk
     -2"]},
{"doc_id": "doc-167",
"chunk_id": ["doc-167-chunk-0", "doc-167-chunk-1"]}
],
"type": "comparison_query",
"results":
{
"topk_chunk_ids": ["doc-492-chunk-0", "doc-176-chunk-0", "doc-168-
     chunk-0", "doc-176-chunk-1", "doc-167-chunk-0"],
"response": "
It appears there might be some confusion in the question as it
     references articles and sports teams that aren't directly aligned
     with the provided context. Let's clarify the information based on
     the chunks provided:

### Australian Rugby (Wallabies)
The provided context does not mention the Davis Cup, which is a tennis
      competition. Instead, it discusses the Australian rugby team,
     known as the Wallabies, and their performance in the Rugby World
     Cup. The Wallabies are facing a critical situation after a poor
     performance in the tournament, particularly after losing to Wales
     40-6. This has put them in a position where they must rely on the
     outcome of a match between Fiji and Portugal to determine their
     fate in the World Cup.

### South African Rugby (Springboks)
The context provided does indicate that the South African national
     rugby team, the Springboks, has achieved significant success and
     improvement in their journey to the Rugby World Cup semi-finals.
     Here are the key points:
1. **2019 World Cup Victory**: Four years ago, the Springboks won the
     World Cup, and they are now preparing to defend their title.
2. **2023 World Cup Performance**: The Springboks have faced
     challenges but have shown resilience and determination. They
     defeated Ireland in a dramatic quarter-final match, 29-28, to reach
      the semi-finals.
3. **Mental and Emotional Preparation**: The Springboks are playing
     with a sense of purpose and for their country, which has driven
     their performance and unity within the team.

### Summary
- **Australian Rugby (Wallabies)**: The Wallabies are not aiming for
     an improvement in a tennis competition (Davis Cup). Instead, they
     are dealing with the consequences of a poor performance in the
     Rugby World Cup and are hoping for a favorable outcome from another
      match to advance.
- **South African Rugby (Springboks)**: The Springboks have indeed
     achieved an improvement in their performance, reaching the Rugby
     World Cup semi-finals despite early setbacks and challenges.

If you have specific articles from 'The Age' and 'Sporting News' that
     you would like to reference, please provide them, and I can offer a
      more detailed comparison."
},
"score": 0
}
```

**QCG-RAG on MultiHop-RAG.** Use Case D.4 shows an example of a query–answer evaluation case by QCG-RAG. From the results, QCG-RAG effectively retrieved queries associated with evidences "`doc-451`" and "`doc-167`", and through query-centric graph reasoning, obtained additional queries linked to "`doc-167`". Based on the retrieved chunk context information, QCG-RAG was able to correctly and effectively identify the comparison entities and contents: "Australia's Davis Cup Team" vs. "South Africa National Rugby Team."

---

**Use Case D.4: QCG-RAG on MultiHop-RAG**

```
{
"question": "Does 'The Age' article suggest that Australia's Davis Cup
    team is aiming for an improvement in their performance compared to
    the previous year, while the 'Sporting News' article indicates
    that the South Africa national rugby team has already achieved an
    improvement to reach the Rugby World Cup semi-finals?",
"answer": "Yes",
"evidence":
[
{"doc_id": "doc-451",
"chunk_id": ["doc-451-chunk-0", "doc-451-chunk-1", "doc-451-chunk
    -2"]},
{"doc_id": "doc-167",
"chunk_id": ["doc-167-chunk-0", "doc-167-chunk-1"]}
],
"type": "comparison_query",
"results":
{
"topk_qids": ["doc-451-chunk-0-16", "doc-167-chunk-0-11", "doc-492-
    chunk-0-17", "doc-407-chunk-0-19", "doc-492-chunk-0-2", "doc-492-
    chunk-0-8", "doc-407-chunk-0-9", "doc-168-chunk-1-7", "doc-440-
    chunk-0-8", "doc-493-chunk-0-1", "doc-407-chunk-1-18", "doc-492-
    chunk-0-7", "doc-167-chunk-1-9", "doc-23-chunk-0-12", "doc-407-
    chunk-0-3"],
"extra_qids_via_knn": ["doc-167-chunk-0-19", "doc-167-chunk-1-15", "
    doc-167-chunk-1-14", "doc-168-chunk-0-12", "doc-168-chunk-1-5", "
    doc-407-chunk-0-16", "doc-407-chunk-0-14", "doc-407-chunk-0-18", "
    doc-407-chunk-1-5", "doc-407-chunk-1-8", "doc-440-chunk-0-9", "doc
    -440-chunk-0-3", "doc-492-chunk-0-12", "doc-493-chunk-0-15", "doc
    -493-chunk-0-0"]
"chunk_to_queries": {"doc-23-chunk-0": ["doc-23-chunk-0-12"], "doc-167
    -chunk-0": ["doc-167-chunk-0-11", "doc-167-chunk-0-19"], "doc-167-
    chunk-1": ["doc-167-chunk-1-9", "doc-167-chunk-1-15", "doc-167-
    chunk-1-14"], "doc-168-chunk-0": ["doc-168-chunk-0-12"], "doc-168-
    chunk-1": ["doc-168-chunk-1-7", "doc-168-chunk-1-5"], "doc-407-
    chunk-0": ["doc-407-chunk-0-3", "doc-407-chunk-0-19", "doc-407-
    chunk-0-9", "doc-407-chunk-0-16", "doc-407-chunk-0-14", "doc-407-
    chunk-0-18"], "doc-407-chunk-1": ["doc-407-chunk-1-18", "doc-407-
    chunk-1-5", "doc-407-chunk-1-8"], "doc-440-chunk-0": ["doc-440-
    chunk-0-8", "doc-440-chunk-0-9", "doc-440-chunk-0-3"], "doc-451-
    chunk-0": ["doc-451-chunk-0-16"], "doc-492-chunk-0": ["doc-492-
    chunk-0-7", "doc-492-chunk-0-8", "doc-492-chunk-0-17", "doc-492-
    chunk-0-12", "doc-492-chunk-0-2"], "doc-493-chunk-0": ["doc-493-
    chunk-0-1", "doc-493-chunk-0-15", "doc-493-chunk-0-0"]},
"sorted_chunks": ["doc-451-chunk-0", "doc-492-chunk-0", "doc-23-chunk
    -0", "doc-407-chunk-0", "doc-167-chunk-0", "doc-407-chunk-1", "doc
    -168-chunk-1", "doc-167-chunk-1", "doc-493-chunk-0", "doc-440-chunk
    -0", "doc-168-chunk-0"],
"topk_chunk_ids": ["doc-451-chunk-0", "doc-492-chunk-0", "doc-23-chunk
    -0", "doc-407-chunk-0", "doc-167-chunk-0"],
"response": "
### Australia's Davis Cup Team Aiming for Improvement
According to the article from 'The Age', Australia's Davis Cup team is
    indeed aiming for an improvement in their performance compared to
```

---

```
        the previous year. Last year, Australia lost the final to Canada.
        This year, they have reached the final again, facing Italy. The
        article highlights the team's journey, noting that they have had a
        relatively easier path to the final, eliminating the Czech Republic
         and Finland, neither of which had top-30 singles players. The
        article also emphasizes the importance of players like Alexei
        Popyrin and the potential for a significant achievement if they can
         secure their 29th Davis Cup title.

    ### South Africa National Rugby Team's Improvement
    The 'Sporting News' article indicates that the South Africa national
        rugby team, known as the Springboks, has already achieved an
        improvement in their performance to reach the Rugby World Cup semi-
        finals. The article details their journey, including a tough loss
        to Ireland in the group stage, which placed them second in Pool B.
        Despite this setback, the Springboks showed their resilience and
        World Cup mettle by narrowly defeating France in a thrilling
        quarter-final match. This victory has put them in a strong position
         to compete for a spot in the final, demonstrating significant
        progress and improvement in their performance.

    ### Summary
    - **Australia's Davis Cup Team**: Aiming to improve on last year's
        performance, which ended in a loss to Canada in the final. They
        have reached the final again and are looking to secure their 29th
        Davis Cup title.
    - **South Africa National Rugby Team**: Has already shown improvement
        by reaching the Rugby World Cup semi-finals, overcoming a tough
        loss to Ireland and a dramatic victory over France in the quarter-
        finals."
    },
    "score": 1
    }
```

