# OpenReview forum: "Query-Centric Graph Retrieval Augmented Generation"
_ICLR.cc/2026/Conference — Submitted to ICLR 2026_

### Official Review · Reviewer_suqq · 2025-10-25

**Soundness:** 3
**Presentation:** 3
**Contribution:** 3
**Rating:** 6
**Confidence:** 4

**Summary:**

This paper presents QCG-RAG, a new graph-RAG framework aimed at solving the "granularity dilemma" in existing methods. The authors' main point is that current methods are stuck in a trade-off: entity-level graphs are too costly and lose context, while document-level graphs miss important details.

The paper's core idea is a "query-centric graph" (QCG) built at an intermediate granularity. They use Doc2Query and Doc2Query to generate query-answer (QA) pairs from text chunks, and these high-quality QA pairs become the nodes in the graph. Edges then connect nodes based on semantic similarity ($E_{intra}$) or link nodes back to their source chunks ($E_{inter}$). The retrieval process then works on this graph, finding relevant query nodes, expanding to neighbors for multi-hop reasoning, and finally gathering the underlying text chunks for the LLM. Experiments on LiHuaWorld and MultiHop-RAG show their method beats other chunk-based and graph-based RAGs.

**Strengths:**

Novel Graph Construction: The main strength is the novel query-centric graph (QCG) construction. Building the graph from synthetic QA pairs is a smart way to get a semantically-rich, intermediate-
granularity index that is "controllable".

Strong Empirical Performance: The method puts up strong numbers, consistently beating baselines on
two different benchmarks. It's especially good on the multi-hop subsets (e.g., 62.12% vs 57.58% on LiHuaWorld multi-hop), which directly backs up the authors' claims about their multi-hop retrieval design.

Thorough Ablation Studies: The ablation studies are comprehensive. The authors did a good job validating their design choices, especially in Figure 4 (showing Query+Answer nodes are best) and in analyzing the impact of all the new hyperparameters (Figures 5, 6).
Clarity and Case Studies: The paper is just very clearly written. The detailed case studies in Appendix D give good qualitative insight into why the method works for multi-hop queries when Naive RAG fails.

**Weaknesses:**

High Construction Cost: The graph construction seems to be expensive. The best results require generating $M=20$ queries per chunk using a 72B model (Qwen2.5-72B-Instruct) The ablation (Figure 4) shows that using a smaller 7B model hurts performance. This reliance on a massive generator model is a big practical drawback for scaling this up to large datasets, as the authors also note.

Hyperparameter Sensitivity: The method introduces a lot of new hyperparameters ($M$, $\alpha$, $h$, $k$, $n$, and $\gamma$) . The ablation plots (Figures 5, 6) show performance can be pretty sensitive to $k$ (neighbors) and $n$ (max nodes). A bigger concern is that the optimal settings were different for the two datasets (e.g., $k=2, n=10, \gamma=1.5$ for LiHuaWorld vs. $k=3, n=15, \gamma=1.0$ for MultiHop-RAG). This suggests that the method might need a lot of careful, dataset-specific tuning, which makes it harder to apply to out-of-domain scenarios.

Ambiguity in Retrieval Similarity: This reviewer finds Section 3.4 a bit unclear on one point. The graph nodes $q$ are
defined as the concatenated $q \oplus a$ pairs . But in Step 1 of retrieval, the paper says it computes $sim(q_u, q)$. Is the user query $q_u$ being compared against the full $q \oplus a$ embedding, or just Novel Graph Construction: The main strength is the novel query-centric graph (QCG) construction. Building the graph from synthetic QA pairs is a smart way to get a semantically-rich, intermediate-
granularity index that is "controllable".

**Questions:**

Q1: To follow up on my point in the weaknesses: in Section 3.4, Step 1, when you compute similarity $s
(q_u, q)$, is $q_u$ being compared to the embedding of the query $q$ alone, or to the embedding of the concatenated query-answer pair $q \oplus a$?

Q2: This reviewer suggests the authors to provide more detail on the construction cost? Specifically, how many total tokens or how
much time did it take to build the graphs for the two datasets using the 72B model? How does this cost compare to the GraphRAG baseline's extraction process?

Q3: The optimal hyperparameters ($k, n, \gamma$) were different across the two datasets. How sensitive is the model to this? What happens if you run the LiHuaWorld experiment using the best parameters from MultiHop-RAG, and vice-versa? This reviewer is curious about how generalizable these settings are.

Q4: Figure 4 is really interesting. "QCG w/ Answer" performs much worse than "QCG w/ Query," and both are worse than the full Query+Answer model. What's your intuition for this? Why are the answers alone so bad as retrieval nodes?

Q5: This reviewer is interested to know the rationale of using these two datasets instead of the commonly adopted QA datasets like 2WikiMQA, HotpotQA, and MuSiQue.

---

### Official Review · Reviewer_A3zU · 2025-10-30

[review text omitted: it was posted to a different submission]

---

### Official Review · Reviewer_7UVr · 2025-10-30

**Soundness:** 2
**Presentation:** 3
**Contribution:** 3
**Rating:** 4
**Confidence:** 5

**Summary:**

This paper presents the QCG-RAG framework, which follows Doc2Query and Doc2Query-- to build a query-centric graph (QCG) and address the granularity issue in graph-based methods, significantly enhancing retrieval for multi-hop reasoning tasks. Experimental results show that QCG-RAG achieves good performance on the LiHuaWorld and MultiHop-RAG datasets.

**Strengths:**

**Good presentation:** This work is easy to read and follow.

**Clear motivation:** The motivation of the QCG-RAG framework is clear, and it proposes an innovative solution for RAG on QCG.

**Weaknesses:**

**Limited Dataset:** The paper only evaluates the model on LiHuaWorld and MultiHop-RAG datasets. While these datasets are useful, they are not sufficient to demonstrate the framework's generalizability across a broader range of tasks. **It is necessary to use more multi-hop reasoning datasets to verify the claim of "enhancing retrieval for multi-hop reasoning tasks" and show the framework's applicability across various contexts.**

**Insufficient Comparison with Multi-Hop RAG Baselines**: Although QCG-RAG performs well in multi-hop reasoning, **the baselines used for comparison are still focused on single-hop scenarios**. To highlight the claimed advantages of QCG-RAG in multi-hop reasoning, it is necessary to include comparisons with other multi-hop RAG methods to demonstrate the superiority of this method in multi-hop reasoning tasks.

**Lack of Cost Analysis:** The paper does not provide an analysis of the computational cost, particularly for large-scale corpora. Given that QCG-RAG involves huge amount of synthetic query generation for all chunks, the computational overhead could be substantial. **It would be good to include a cost analysis and compare it with the baseline methods, especially under different corpus sizes. Without this comparison, discussing effectiveness alone is somewhat unfair, and the practical value of the method remains unclear.**

**Questions:**

The quality of queries generated by different models — as well as the varying preferences of different retrievers toward these queries — would intuitively have a significant impact on the proposed method. I believe this is a necessary analysis with important practical and methodological implications.

---

### Official Review · Reviewer_z6eu · 2025-10-30

**Soundness:** 2
**Presentation:** 3
**Contribution:** 2
**Rating:** 2
**Confidence:** 4

**Summary:**

This paper aims to address the "granularity dilemma" in graph-based RAG methods, and proposes Query-Centric Graph Retrieval-Augmented Generation (QCG-RAG). QCG-RAG uses extended Doc2Query techniques to query-answer pairs as augmented query graph for enhancing retrieval. Retrieval involves multi-hop expansion over this graph to identify relevant chunks.

Experiments on two datasets (MultiHop-RAG, Lihuaworld)  claim superior accuracy over baselines like graphRAG and NaiveRAG.

**Strengths:**

1. Well motivated.

The granularity trade-off in graph-RAG is a timely issue, as highlighted in recent works (e.g., GraphRAG). The paper correctly identifies limitations like high token costs in entity extraction and semantic loss in coarse graphs. These are practical challenges when applying graphRAG algorithms in realworld scenarios.

2. Clear Presentation.

The introduction and methodology sections are structured logically, with formal definitions and a helpful overview figure. Ablations on node formulations provide some insight into design choices.

**Weaknesses:**

1. Limited Novelty.

The core contribution *using Doc2Query-generated queries as graph nodes* is not sufficiently original for ICLR. Doc2Query and Doc2Query-- are established document expansion techniques, and their integration with graphs feels like a straightforward, engineered combination rather than a breakthrough. The "query-centric" framing is appealing but overstates the innovation; it's essentially a hybrid index, not a "new paradigm" as claimed.

2. Technical Defects.

- The similarity function in retrieval (s(qu, q) = sim(qu, q) + ε, with ε=1) is ad-hoc and lacks motivation—why add a constant bias?
- The graph construction uses KNN for edges but doesn't discuss scalability (e.g., for large corpora, O(N^2) similarity computations could be prohibitive).
- Efficiency analysis are not give, what is the complexity (token cost, indexing time, retrieval time)of the algorithm on large algorithm?
- Hyperparameters (e.g., α=80%, h=1, k=2-3, γ=1.0-1.5) are dataset-specific and tuned aggressively, potentially overfitting to benchmarks.
- False statement or clarity issue: (1) the begining of the introduction (line 26) "RAG has become a standard approach for improving the factuality of LLMs....." is not true, there are many ways, e.g. fintuning, or knowledge editing. (2)  "Controllable granularity" is overused without precise definition.


3. Empirical Shortcomings.

- Datasets are limited to two (LiHuaWorld, MultiHop-RAG). While Lihuaworld is not a widely-used datasets. Other popular RAG datasets such as Musique and HotpotQA are not included. Plz see HippoRAG[1] for the dataset details.
- Baselines are cherry-picked. Only naive RAG and several GraphRAG with poor performance (comparable to or worse than Naive RAG) are included. All state-of-the-art graphRAG algorithms (HippoRAG[1], HippoRAG2[2], GFM[3], Raptor[4], E^2GraphRAG[5], LogicRAG[6]) are missing.
- No efficiency validation is given.
- No idea how the performance will be when the corpora become large. The edges are built based on similarity, so the *similary queries* can become overwhelming and densely connected, which can be disaster for the graph-expansion based retrieval: the recalled passages can be overloaded and noisy.






---
References

- [1] HippoRAG: Neurobiologically Inspired Long-Term Memory for Large Language Models, Neurips'24
- [2] From RAG to Memory: Non-Parametric Continual Learning for Large Language Models NeurIPS'25
- [3] GFM-RAG: Graph Foundation Model for Retrieval Augmented Generation, NeurIPS'25
- [4] RAPTOR: Recursive Abstractive Processing for Tree-Organized Retrieval, neurips'24
- [5] E^2GraphRAG: Streamlining Graph-based RAG for High Efficiency and Effectiveness, arxiv'25
- [6] You Don't Need Pre-built Graphs for RAG: Retrieval Augmented Generation with Adaptive Reasoning Structures, arxiv'25

**Questions:**

How does your algorithm work on other datasets and comparing with other baselines?

How does the algorithm perform on large datasets?

How do you control the precision and recall of the retrieved passages with theoretically bounded error rates.

---

### Meta-Review · Area_Chair_ZuSw · 2026-01-06

**Summary:**

This paper studies graph-based RAG for enhancing long-context understanding and multi-hop reasoning by exploring an optimal trade-off between cost and performance through granularity control. The paper initially received overall mixed and slightly negative ratings, with one reject (rating 2), one marginally below the acceptance threshold (rating 4), and two marginally above the acceptance threshold (rating 6). Reviewers acknowledge several strengths such as the clear motivation, presentation quality, and empirical gains. However, reviewers raised the following major concerns:

1. The absence of theoretical analysis
2. The missing analyses for the sensitivity to multiple hyperparameter choices
3. The need for evaluation on more diverse datasets
4. High graph construction costs
5. Insufficient clarification on technical details, leading to numerous questions

**Reviewer Concerns:**

The authors did not provide any rebuttal. Therefore, none of the above concerns was addressed.

**Reviewer Scores:**

The initial ratings would not change, since no rebuttal has been provided.

---

### Decision · Program_Chairs · 2026-01-26

Reject